# Effect of Acute Self-Myofascial Release on Pain and Exercise Performance for Cycling Club Members with Iliotibial Band Friction Syndrome

**DOI:** 10.3390/ijerph192315993

**Published:** 2022-11-30

**Authors:** Jong Jin Park, Hae Sung Lee, Jong-Hee Kim

**Affiliations:** 1GYMNOW Fitness, Seoul 04417, Republic of Korea; 2Department of Physical Education, College of Performing Arts and Sport, Hanyang University, Seoul 04763, Republic of Korea; 3BK21 FOUR Human-Tech Convergence Program, Hanyang University, Seoul 04763, Republic of Korea

**Keywords:** cycling, self-myofascial release, iliotibial band friction syndrome, pain, cycling performance

## Abstract

Cycling is a popular sport, and the cycling population and prevalence of related injuries and diseases increase simultaneously. Iliotibial band friction syndrome is a common chronic overuse injury caused by repetitive knee use in cycling. Self-myofascial release using foam rollers is an effective intervention for this syndrome; however, studies reporting positive results on self-myofascial release in cycling are limited. Therefore, this study investigated the effect of self-myofascial release on pain and iliotibial band flexibility, heart rate, and exercise performance (cadence, power, and record) in adult male cyclists with iliotibial band friction syndrome. We evaluated the pain and exercise ability of the control (*n* = 11) and self-myofascial release (*n* = 11) groups before and after cycling twice. Significant differences were observed in the pain scale, the iliotibial band flexibility, and cycling pain and power. The posterior cadence of the self-myofascial release group was 3.2% higher than that of the control group. The control group’s record time increased by 74.64 s in the second cycling session compared to the first cycling session, while that of the self-myofascial release group decreased by 30.91 s in the second cycling session compared to the first cycling session. Self-myofascial release is effective in relieving pain and may improve cycling performance by increasing the iliotibial band flexibility.

## 1. Introduction

A bicycle is an eco-friendly, convenient means of transportation, and cycling, as a sport, significantly benefits the physical and mental health of participants. In addition, despite the relatively simple design of bicycles, achieving a high speed is possible, increasing the motivation to participate in the exercise and, thus, increasing physical activity. Recently, given the increased awareness of the global eco-friendliness of cycling [1], decrease in participation in indoor sports (such as weight training and swimming), and increased participation in outdoor sports (such as walking and cycling), due to variants of the SARS-CoV-2 pandemic [2], interest in cycling is expected to continue. Following this trend, the number of cycling participants is currently reported to reach 2 billion worldwide, and is expected to rise to 5 billion by 2050 [3]. All cycling-cause injury rate who engage in competitive cycling exercise, including cycling club members, was about 30.1% [4].

The motion in the impedance zone when increasing power during pedaling of a bicycle is a pedal-pushing motion, with the knee joint extended at 90–110°, and the opposite knee joint flexed at 30–35°, with repeated rotation [5]. During this movement, the quadriceps femoris, gastrocnemius, soleus, and gluteus maximus are involved in extension. Regarding flexion, the rectus femoris, hamstring, iliopsoas, and tibialis anterior generate power [6]. In general, a skilled person with a cadence of 85 rotations per min (RPM) repeats approximately 5000 pedals in 1 h, which can lead to severe injuries due to a fine range of motion (ROM) restrictions, anatomical structure, and posture imbalance [7]. In order to prevent these injuries and perform exercises smoothly, mobility and flexibility of connective tissues, such as the muscles, ligaments, tendons, and fascia surrounding the joints, must be secured. Moreover, if mobility and flexibility are limited, inflammatory vasoactive neuropeptides are released with severe pain [7]. If this pain is overlooked and the individual does not rest and undergo rehabilitation, it can lead to serious chronic conditions, such as tendinosis and iliotibial band friction syndrome (ITBFS) [8]. ITBFS frequently occurs in people who participate in sports involving repetitive knee use, such as cycling, running, skiing, soccer, basketball, and tennis [9,10]. ITBFS, accompanied by a sharp stabbing pain in the outer knee area, is reported in 12% of runners [11] and 15–24% of cyclists [12]. When pedaling a bicycle with a limited joint ROM, the rapidly repeated flexion and extension of the knee puts excessive fatigue and pressure on the iliotibial band (ITB) on the upper lateral femoral epicondyle. Consequently, it causes microscopic damage to the knee tendons, bones, and fascia, resulting in inflammatory pain [13,14]. ITBFS pain is so severe that even elite cyclists with top-level skills give up training and events or experience poor performance [13].

Non-invasive rehabilitation tools, such as stretching, massage, and fascial relaxation, are recommended to effectively relax local tissue adhesions through repeated friction [15]. Particularly, due to ITB division into the proximal, median, and distal parts and physical elongation, it is not feasible to expect effective ITB elongation only with clinical stretching. It has been suggested that self-myofascial release (SMR), using a foam roller, can be more effective [16]. SMR using a foam roller can be used immediately on the painful area regardless of time and place. Furthermore, SMR improves fascial damaged tissue adhesion, blood circulation, autonomic nervous system control, ROM, and flexibility and reduces pain [17,18]. Enthusiasts and club members who actively participate in exercise tend to be highly motivated to exercise and have strong will to return to exercise or sport, so they overlook the pain and are less likely to actively participate in long-term rehabilitation programs. Studies have been recently conducted to confirm the immediate effect of SMR; however, there are conflicting opinions on the impact so far [19,20,21,22]. It has also been suggested that ITBFS has complex and unpredictable mechanisms, typically following a fluctuating course, with relapse or improvement at any point in the treatment progression [23]. Furthermore, emerging studies are establishing optimal SMR programs and verifying that they can induce immediate benefits. However, since ITBFS is clinically diagnosed, patient recruitment, research design and application, and ethical issues are major barriers to research. Until recently, related studies analyzed exercise performance (such as ROM, flexibility, and jump power) at a basic level, and exercise performance in actual sports was unclear [24]. Therefore, it is necessary to establish an optimal acute SMR ITBFS program and verify the effectiveness of exercise for effective recovery and rapid return to the field among club members with high motivation to participate in cycling.

This study aimed to investigate the effect of one-time SMR using a foam roller, an effective method of mediating ITBFS, via special tests, visual analog scale (VAS), and exercise performance on adult male cycling club members diagnosed with ITBFS. Hence, we hypothesized that the SMR group would show positive pain relief and exercise performance compared to the control group. The participants were divided into the control and SMR groups, and the differences between VAS, heart rate (HR), exercise performance (cadence, power, record), and special test results (Renne’s test, Noble’s compression test, Ober’s test) while cycling, and after acute SMR, were compared and analyzed.

## 2. Materials and Methods

### 2.1. Study Subject

This study was conducted on informed cycling club members who consented after knowing the purpose and method of this study, in Seoul, Korea. The inclusion criteria were: (1) Adult men aged 20–45 y, (2) Individuals with over one year of cycling experience who had been active for the past year, (3) Those who had experienced knee pain while pedaling a bicycle, (4) Individuals positive in the ITBFS diagnostic test. The exclusion criteria were: (1) Individuals in poor condition due to drinking and lack of sleep, (2) Those whose joint surgery history might affect the study results, and (3) Individuals who did not consent to the study. Finally, 22 candidates were enrolled in the study, excluding two with negative ITBFS test results. They were grouped using the convenience sampling method (Table 1).

### 2.2. Study Design

ITB flexibility and pain scale were measured through a pre-special ITFBS positive test in the control and SMR groups. After the first 10 km cycling course, the control group had a static rest for 120 min [25]. The SMR group conducted the intervention using a foam roller for 20 min after a static rest of 100 min, and then both groups underwent a post-cycling special test after the second cycling on the same course. The exercise was evaluated by collecting the HR, cadence, power, and record results during the first and second cycling sessions (Figure 1).

#### 2.2.1. Special Test

Using the results of previous studies, Renne’s test and Noble’s compression test for VAS assessment, and Ober’s test for ITB flexibility assessment, were performed as diagnostic methods for ITBFS (Table 2) [26,27]. In addition, a digital angle meter (AG-02LB, Gain Express Holdings Ltd., Kwawan, Hong Kong) was used to obtain accurate results. An experienced licensed physical therapist with over five years of clinical experience conducted the special tests. The tests were performed pre-first and post-second cycling.

#### 2.2.2. Visual Analog Scale to Self-Evaluate the Pain

VAS was conducted to self-evaluate the pain felt while cycling after the 1st cycling and the 2nd cycling. VAS required the degree of pain felt during the first and second cycling to be directly recorded immediately after each cycling session. The range of 1–10 was used, with 0 representing absence of pain and 10 extreme pain.

#### 2.2.3. Bicycle and Exercise Performance

Zwift (Zwift Inc., Long Beach, CA, USA), a bicycle simulation software, and a roller fixture (Tacx NEO T2 Smart Trainer, Garmin Korea Ltd., Seoul, Republic of Korea) were used to eliminate external factors that might affect the research results, such as weather, wind, traffic, and road conditions, and to provide the same cycling environment [28,29]. The cumulative altitude per 10 km cycling was 62 m, the longest uphill section was 1.5 km, and the cumulative altitude was 18 m. The participants cycled 10 km each during the first and second cycling sessions. Cadence, power, and record results were collected per second via Zwift to analyze their performance. An HR measuring device (Polar H-10, Polar Electro Inc., Kempele, Finland) was used for HR analysis [30].

#### 2.2.4. Self-Myofascial Release

The SMR applied in a previous study [31] was modified and supplemented to fit the purpose of this study. It was constructed for the triceps surae, tibialis anterior, quadriceps femoris, tensor fasciae latae, and gluteus maximus (Table 3). A hard type 66 × 14 cm foam roller, made of Ethylene Vinyl Acetate Copolymer, was used as the SMR tool.

### 2.3. Statistical Analyses

All the data collected during this study are expressed as mean ± SEM. The statistical analyses were performed using GraphPad Prism 9 (GraphPad Software, Inc., San Diego, CA, USA). Two-way analysis of variance (ANOVA) was used to examine the effects of SMR on 1st cycling and 2nd cycling during the experimental period. A Tukey HSD post-hoc test was used to compare the mean differences in the special tests (Renne’s test, Noble’s compression test, Ober’s test, VAS) and exercise performance (heart rate, cadence, power, and record) between the control and SMR groups. Statistical significance was set at *p* < 0.05.

## 3. Results

### 3.1. Visual Analog Scale and Iliotibial Band Flexibility

Comparative analysis of the difference between VAS felt while cycling, and ITB flexibility confirmed in the special test, revealed that no significant difference was observed in Renne’s VAS test results between control (*p* = 0.9352) and SMR group (*p* = 0.9982) in each cycling session. Renne’s test results for pre-1st and post-2nd cycling of the control showed no difference (*p* = 0.9912), and the SMR group showed a slight decrease after SMR application, but there was no significant difference (Figure 2A; *p* = 0.2860). In the case of VAS through Nobel’s compression test, the result of the SMR pre-1st cycling was significantly higher than that of the control pre-1st cycling (*p* = 0.0088), the control post-2nd cycling (*p* = 0.0023), and the SMR post-2nd cycling (Figure 2B; *p* < 0.0001). No significant difference was observed in ITB flexibility in pre-1st cycling through Ober’s test between each group (*p* = 0.9917). However, the ITB flexibility of the SMR group significantly increased post-2nd cycling and was more than that observed in pre-1st cycling (Figure 2C; *p* < 0.05). Regarding the VAS of pain felt while cycling, no significant difference was observed between the two groups after the 1st cycling (*p* = 0.8835). However, the SMR group’s VAS of pain significantly decreased compared to those of the post-1st cycling and the post-2nd cycling of the control (*p* < 0.01). In addition, the SMR group experienced a more significant decrease in the 2nd cycling than in the 1st cycling (Figure 2D; *p* < 0.0001).

### 3.2. Exercise Performance

HR results presented the physiological response of the heart according to exercise intensity. The post-2nd cycling HR of the SMR group was 150.3 ± 5.9, which was rather high compared to the results observed after 1st cycling (132.4 ± 5.9) and 2nd cycling (138.9 ± 6.1) of the control and the SMR 1st cycling (141.1 ± 8.9); however, no significant difference was observed. Regarding cadence, in which the number of pedals per min was measured, the 2nd cycling of the SMR group was 83.1 ± 3.1, which was slightly higher than that observed after 1st cycling (80.2 ± 2.4) and 2nd cycling (81.1 ± 2.8) of the control, and the SMR 1st cycling (82.4 ± 3.3); however, no significant difference was observed (Figure 3B). On analyzing the pedaling power while cycling, no significant difference was observed between the groups in the 1st cycling results; however, the SMR group displayed significantly higher pedaling power than the control group in the 2nd cycling results (103.4 ± 8.7 vs. 140.3 ± 8.9; *p* < 0.01; Figure 3C). The 2nd cycling record time of the SMR group was 1320.7 ± 48.2 s for 10 km cycling, which was slightly decreased compared to that observed among the control 1st cycling (1472.5 ± 92.8), the control 2nd cycling (1547.2 ± 98.5), and the SMR 1st cycling (1351.2 ± 47.1); however, no significant difference was observed. On the other hand, on analyzing each section of the longest uphill road (distance: 1.5 km; cumulative altitude: 18 m) in the 2nd cycling, a significant difference was observed between the groups in the 500 m, 600 m, 900 m, 1000 m, 1200 m sections, and the average value of the accumulated power in each section also significantly differed between the groups (Table 4).

## 4. Discussion

This study aimed to investigate the effect of one-time SMR on pain and exercise performance in individuals diagnosed with ITBFS who cycled regularly over one year among adult male cycling club members. We found that the one-time SMR program was effective in relieving pain and improving motor performance by enhancing VAS, ITB flexibility, and pedaling power during cycling. In addition, we tried to control other environmental factors that could affect the study results using the latest smart sports software and equipment. No participant complained of side effects or suffered injuries during the study’s special tests, cycling, and SMR.

Among the special tests in this study, Renne’s test showed no difference between groups in pre-1st cycling, so it was possible to clearly evaluate the SMR effect after the second cycling. On the other hand, in Noble’s compression test, a significant difference was observed in the pre-cycling results between both groups; therefore, it was difficult to clearly interpret the effect of SMR in the results after the second cycling. Due to the nature of the special test, direct stress (such as elevation, pressure, and palpation) was applied to the damaged tissue in a specific posture vulnerable to pain; therefore, it was not feasible to expect pain relief from a one-time SMR in Noble’s compression test. There was no significant difference between groups in the VAS measured after the first cycling due to appropriate pre-cycling control; however, significant differences within and between groups were confirmed in the SMR group’s VAS after the second cycling. Therefore, SMR was effective in relieving pain while cycling. Physical compression by SMR using a foam roller mediates the sensitivity of a nociceptor that transmits pain to the central nervous system and a mechanoreceptor that detects physical deformation, such as pressure, elongation, and flexion. SMR relieves pain by inducing the rearrangement of muscles, ligaments, tendons, and fascia through artery expansion and increased blood flow at the injured site [32,33]. Particularly, the sensory neurons of the fascia are more densely innervated than other tissues and, thus, play an essential role in determining the degree of pain. However, strong stimulation is required to effectively relax the fascia because it has to penetrate the thick skin and muscle resistance. SMR using a foam roller can apply a pressure of approximately 7 VAS [32]; therefore, the SMR applied in this study would have relaxed the adhesion of the fascia to the damaged area. The mechanism of diffuse noxious inhibitory control (DNIC) might be related to these results [34,35]. DNIC is a phenomenon in which monoamines, such as noradrenaline and serotonin, are expressed by specific stimuli to counter-irritate pain signals from the spinal cord to the brain [32]. This may temporarily relieve the pain caused by ITBFS [34,35]. Furthermore, this study’s SMR was performed for 20 min, which was longer than the SMR performed in previous studies (30 s–15 min) [21,32,36]. This suggests the applicability of the DNIC mechanism, as it delivered high physical pressure to the SMR group. 

On the other hand, this pain relief mechanism might be closely associated with the increased ITB flexibility identified in the SMR group [31,37]. ITB originates from the gluteus maximus proximal outside the hip joint and is structured such that the bifurcated fascia, the “iliotibial tract”, crosses the musculus biceps femoris and vastus lateralis and connects the lateral Gerdy’s tubercle of the tibia [38]. ITB has a thick structure in a complex combination with the tensor fasciae latae in the gluteus maximus proximal; therefore, it is not feasible to expect the optimal effect from traditional stretching alone [16]. Thus, to secure flexibility through ITB elongation, SMR application using a foam roller is recommended. This study confirmed a significant difference within the SMR group in investigating ITB flexibility through Ober’s test. ROM restriction, due to decreased flexibility, induces an imbalance of muscles, ligaments, tendons, and fascia surrounding the joint, leading to stiffness, adhesions, trauma, inflammation, and decreased venous and lymphatic system function between connective tissues [39]. Consequently, motor function decreases in efficiency, leading to various musculoskeletal injuries and diseases. Many studies on sports have conducted research on various stretching exercises to increase athletes’ ROM; previous studies have reported that static stretching can negatively affect athletic performance temporarily [40,41,42,43,44]. SMR using a foam roller does not affect muscle strength and is, therefore, recommended as an effective intervention method that can increase ROM [45].

However, conflicting results were reported in related studies. Richman et al. reported that the effect of SMR on flexibility was not significant when comparing 6-min SMR and flexibility in dynamic stretching in adult women [46]. Conversely, Bradbury-Squires et al. reported that the flexibility of the SMR group increased by 16% compared to the control group after 5 min of SMR in adult men [47]. SMR studies using foam rollers have significant limitations compared to previous studies due to the demographic characteristics of participants and the diversity of sports events, environments, tools, and SMR programs. However, compared to previous studies mentioned above, the different delivery of the foam roller’s physical pressure, according to weight differences (69.3 ± 10.9 vs. 84.4 ± 8.8 kg) based on the participant’s sex, is essential to determining the results of both studies. In addition, SMR application time is also important. In previous studies, 30 s of SMR application had no significant effect on flexibility and exercise capacity [48,49]; nonetheless, a 90-s SMR study showed improved flexibility [22,50]. In particular, Monteiro et al. suggested that ‘more than 90 s’ could be a criterion for inducing improved flexibility with SMR, and establishing a systematic SMR program with extended time was required to obtain a better effect [51]. The mean body weight of the SMR group in this study was 77.2 ± 7.7 kg, and SMR was performed for a longer time (20 min) than in other previous studies. Therefore, the ITB flexibility increase observed in this study was due to the SMR program, consisting of a large body weight and a relatively long time, and this increase in ITB flexibility might affect cycling athletic performance [52].

In the cadence, power, and record analyzed as motor ability variables of cycling in this study, no significant difference was observed in the variables, excluding power. However, since a minimal difference of <1% or 1 s in cycling capacity is essential to determine race outcome [52,53], we could reach some conclusions to guide follow-up studies. Cadence is the number of pedals per unit of time, increasing with high power output [54]. A previous study suggested 3.4–5.5% as a significant difference (*p* < 0.05) determining cycling performance [55]. In this study, while the cadence between both cycles of the control group increased by 0.4% (1st cycling: 80.2 ± 2.4 vs. 2nd cycling: 80.5 ± 2.6 RPM), in the SMR group, the cadence increased by 1.3% (1st cycling: 82.0 ± 3.1 vs. 2nd cycling: 83.1 ± 3.1 RPM) in the second cycle compared to the first. When this was compared with the 2nd cycling results between groups, the cadence of the SMR group increased by 3.2% compared to the control group (*p* = 0.9201). While cycling, ITBFS caused more pain on inclines than on leveled ground and lowered exercise capacity; therefore, power was analyzed by dividing the values into units of 100 m during the longest uphill section in the 2nd cycling. Consequently, the power of the SMR group was significantly higher than that of the control group in the 500 m, 600 m, 900 m, 1000 m, and 1200 m sections. A significant difference was observed between the groups in the cumulative average results. The power displayed an increasing pattern in the SMR group and a decreasing pattern in the control group from the 300 m section. There were some limitations in scientific interpretation because the statistical differences were not confirmed in all the sections. However, there were some interesting previous results [56] that confirmed the maximum torque increase in the quadriceps and hamstrings by applying SMR to cross-country skiers with varying inclinations of the cycling course. The longest significant difference in some sections of the uphill road resulted from improved exercise capacity due to SMR. Overall, the increased cadence and power confirmed in the SMR group also affected the record time(s). No significant difference was observed between the groups as regards the record time; however, the 2nd cycling record time increased by 74.64 s compared to the 1st cycling in the control group. In contrast, the 2nd cycling record decreased by 30.91 s compared to the 1st cycling in the SMR group. Reducing pain in ITBFS using SMR could reduce the burden of bicycle pedaling and offer a quick return to pre-ITBFS levels [57]. ITBFS pain is likely to reduce with repeated knee movement at a higher speed than at a slow speed [57]. In elite table tennis players, dynamic stretching and self-myofascial release using a foam roller significantly improved flexibility, power, ball speed, and agility. It was suggested that the use of these protocols could improve performance [58]. In addition, other studies reported a decrease in pain perception after acute rolling massage [59,60]. It was reported that foam rolling reduced pain perception for exercise-induced delayed onset of muscle soreness recovery, which resulted in decreased pain perception by restoring soft tissue extensibility and/or activating the central pain control system [32]. However, more studies are needed on the mechanisms and interaction involved in SMR and pain relief.

A limitation of this study was that it was conducted on adult male cycling club members diagnosed with ITBFS, so there was a limit to generalizing all the results. However, in the future, interesting results can be drawn if a follow-up study on pain and exercise performance is conducted on other populations, such as adolescents, the elderly, women, and other athletes, while considering the application time, duration, and diversity of SMR applications. In addition, it will be necessary to add data by recruiting more subjects to secure the high reliability of the results of this study.

## 5. Conclusions

This study aimed to verify the effects of special tests, VAS, and exercise ability based on one-time SMR in adult male cycling club members with ITBFS. Significant differences were observed in the SMR group in VAS through Nobel’s compression test, ITB flexibility through Ober’s test, and VAS and power while cycling. No significant difference was observed in HR, cadence, and record time; however, the post-cycling cadence of the SMR group increased by 3.2% compared to the control group, and the record time decreased by 30.91 s. Considering that a minimal difference of 1% and 1 s can determine cycling ability and results, one-time SMR relieved pain and improved the exercise ability of cyclists suffering ITBFS. Considering the confirmative effect of SMR in the current study, it is expected that a systematic SMR program consisting of at least 4 min for each body part and a total of 20 min or more would help relieve pain and improve exercise performance when applied to relevant field and clinical practice situations.

## Figures and Tables

**Figure 1 ijerph-19-15993-f001:**
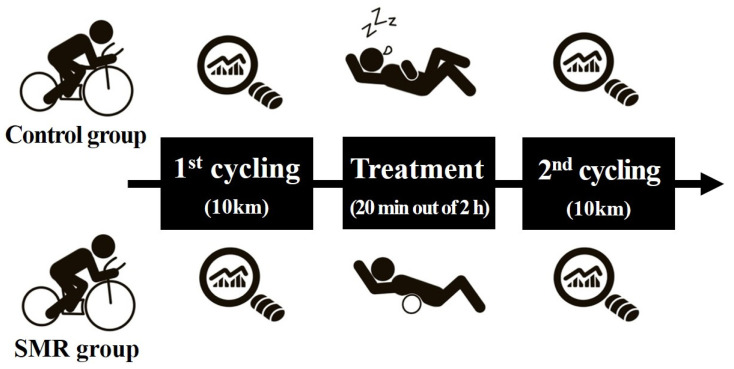
Schematic design of experimental procedures.

**Figure 2 ijerph-19-15993-f002:**
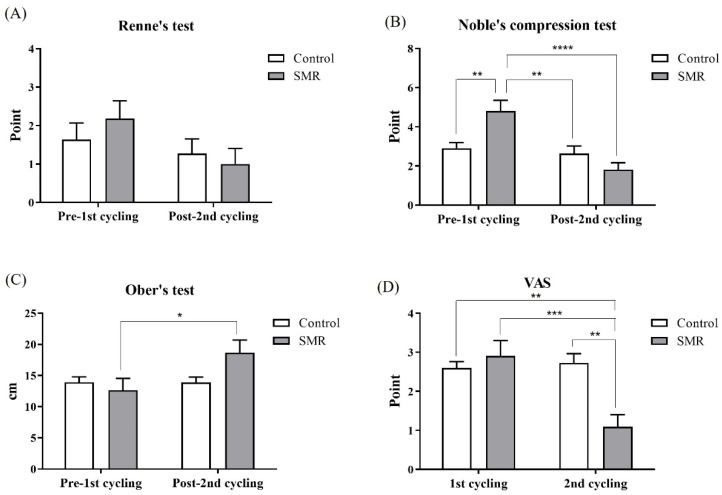
Comparison of iliotibial band flexibility and visual analog scale. Comparison of iliotibial band flexibility and visual analog scale. (**A**) Results of the Renne’s test; (**B**) Results of the Noble’s compression test; (**C**) Results of the Ober’s test; (**D**) Results of the visual analog scale test. * Superscripts denote statistically significant values (* *p* < 0.05; ** *p* < 0.01; *** *p* < 0.001; **** *p* < 0.0001).

**Figure 3 ijerph-19-15993-f003:**
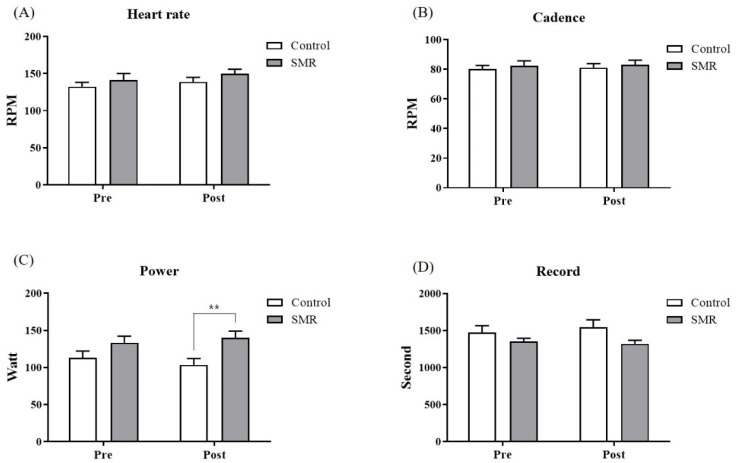
Comparison of exercise performance. (**A**) Results of the Heart rate; (**B**) Results of the Cadence; (**C**) Results of the Power; (**D**) Results of the Record. ** Superscripts denote statistically significant values (** *p* < 0.01).

**Table 1 ijerph-19-15993-t001:** Characteristics of study subjects.

Group	Sex	Age (year)	Height (cm)	Weight (kg)	Career (year)
Control group (*n* = 11)	Male	32.8 ± 4.8	174.3 ± 4.3	75.1 ± 6.0	4.1 ± 2.0
SRM group (*n* = 11)	Male	34.8 ± 4.3	175.8 ± 4.2	77.2 ± 7.7	3.5 ± 1.8

All data are represented as mean ± SEM.

**Table 2 ijerph-19-15993-t002:** Special test method for evaluation of iliotibial band friction syndrome.

Special Test	Method
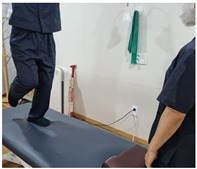	[Renne’s test] 1. The subject sat down until knee was at flexion 30° stand up. 2. Pain in the lateral epicondyle was checked. * Positive on pain 3. Subjective visual analog scale evaluation was conducted
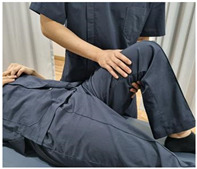	[Noble’s compression test] 1. The subject’s knee was passively flexed to 90°. 2. The examiner applied pressure on lateral epicondyle and the upper gradually. 3. The knee was self-extended and checked for lateral pain before 30°. * Positive on pain 4. Subjective visual analog scale evaluation was conducted.
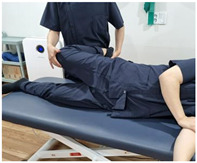	[Ober’s test] 1. The subject lay in the lateral position with the test leg up. 2. The subject’s knee was flexed at 90° and the ankle and knee supported. 3. When the support was released, the degree of abduction (cm) of the knee was measured *. * Negative for adduction, positive for abduction

**Table 3 ijerph-19-15993-t003:** Self-myofascial release programs.

Target Muscle	Self-Myofascial Release	Protocol	Frequency	Time
Triceps surae	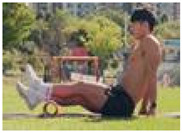	1. Apply forward and backward at the trigger point 2. Apply left and right at the trigger point 3. Apply left and right after ankle rotation at the trigger point 4. Repeat at the downward, middle, and, upward positions of the triceps surae	1. 4 times 2. 4 times 3. 4 times 4. 1 time	4 min
Tibialis anterior	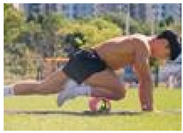	1. Apply forward and backward at the trigger point 2. Apply left and right at the trigger point 3. Apply left and right after ankle flexion and extension at the trigger point 4. Repeat at the downward, middle, and, upward positions of the tibialis anterior	1. 4 times 2. 4 times 3. 4 times 4. 1 time	4 min
Quadriceps femoris	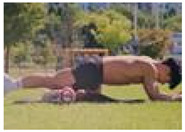	1. Apply forward and backward at the trigger point 2. Apply left and right at the trigger point 3. Apply left and right after knee flexion and extension at the trigger point 4. Repeat at the downward, middle, and, upward positions of the quadriceps femoris	1. 4 times 2. 4 times 3. 4 times 4. 1 time	4 min
Tensor fasciae latae	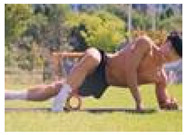	1. Apply forward and backward at the trigger point 2. Apply left and right at the trigger point 3. Apply left and right after knee flexion and extension at the trigger point 4. Repeat at the downward, middle, and, upward positions of the tensor fasciae latae	1. 4 times 2. 4 times 3. 4 times 4. 1 time	4 min
Gluteus maximus	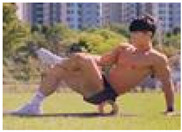	1. Apply forward and backward at the trigger point 2. Apply left and right at the trigger point 3. Apply after abduction and adduction of the hip joint at the trigger point 4. Apply after flexion and extension of the hip joint at the trigger point	1. 4 times 2. 4 times 3. 4 times 4. 4 times	4 min

**Table 4 ijerph-19-15993-t004:** Power in the longest uphill section.

Distance (m)	Power (watt)		*p*
	Control	SMR	
100	106.8 ± 12.9	116.5 ± 8.5	ns
200	109.6 ± 8.9	123.2 ± 8.5	ns
300	114.8 ± 6.9	138.2 ± 8.3	ns
400	110.4 ± 6.8	139.0 ± 9.0	ns
500	105.9 ± 6.6	146.1 ± 8.5 **	<0.01
600	108.8 ± 7.7	144.8 ± 6.8 *	<0.05
700	109.1 ± 7.5	138.5 ± 6.0	ns
800	104.0 ± 6.1	133.5 ± 8.6	ns
900	96.0 ± 6.2	135.6 ± 9.4 *	<0.05
1000	97.9 ± 6.3	138.5 ± 7.5 **	<0.01
1100	104.0 ± 8.9	132.2 ± 8.1	ns
1200	95.8 ± 6.1	136.3 ± 7.4 **	<0.01
1300	102.9 ± 7.8	135.2 ± 7.0	ns
1400	104.6 ± 6.6	137.7 ± 7.3	ns
1500	105.0 ± 7.4	137.2 ± 6.3	ns
Mean power	105.0 ± 5.2	135.5 ± 7.2 *	<0.05

All data are represented as mean ± SEM. * Superscripts denote statistically significant values (* *p* < 0.05; ** *p* < 0.01).

## Data Availability

All data used to support the findings of this study are included in the article. The analyzed data during the current study are available from the corresponding author upon reasonable request.

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
