# Peer review of "Effect of Acute Self-Myofascial Release on Pain and Exercise Performance for Cycling Club Members with Iliotibial Band Friction Syndrome"

_ijerph, 2022, doi:10.3390/ijerph192315993_

Round 1
Reviewer 1 Report
Hello,
Your article is interesting and of use to many in the cycling and sport community. The use of SMR is popular yet we are still lacking evidence for it's use. To improve this study see the comments below:
Line 14 – common cycling “disease” – usually not referred to in this way. Chronic overuse injury more likely to be used.
Line 38 – can you provide some numbers here? As well as a reference.
LINE 89-91 – In the hypothesis be more clear on your direction of results. What do you mean by the SMR group would be positive compared to the control group?
I look forward to your revisions.
Thank you.
Author Response
Dear Reviewer_1
Thanks for your constructive comments, and it contributed a lot to completing a more advanced manuscript. We will write a response to your comments and attach it, so please refer to it.
kind regards,
Dr. Jong-Hee Kim

Reviewer 2 Report
Dear authors,
I had the opportunity to review the paper entitled “Effect of acute self-myofascial release on pain and exercise performance for cycling club members with iliotibial band friction syndrome”. After reading the manuscript I have some concerns that should be resolved:
Abstract: Please add reference to the sentence for line 21-22.
Methods: How did you calculate the sample size?
Statistical analyses: Is it SEM or SD?
Results: Please add p value to line 163-164
Results: Please give exact p values
Results: Did you record the adverse events? You mentioned aob
Discussion: Please briefly summarize the results in the first paragraph of the discussion.
Discussion: inclusion criteria: “2) Individuals with over one year of cycling experience who have been active for the past year 4) Individuals positive in the ITBFS diagnostic test.”. But your first sentence is “diagnosed with ITBFS for one year” please revise this sentence.
Discussion: Did you add this article to the discussion section? “Immediate Effect of Pressure Pain Threshold and Flexibility in Tensor Fascia Latae and Iliotibial Band According to Various Foam Roller Exercise Methods. 2019”
Discussion: What are your recommendations to the clinicians?
Discussion: Please add a limitation section to this part.
Author Response
Dear Reviewer_2
Thanks for your constructive comments, and it contributed a lot to completing a more advanced manuscript. We will write a response to your comments and attach it, so please refer to it.
kind regards,
Dr. Jong-Hee Kim

Reviewer 3 Report
The manuscript “Effects of acute …” reports the findings from a study examining the impact of self-myofascial release on pain generated by iliotibial band friction among bicyclists, a population susceptible to tis malady. The research team assessed the efficacy of myofascial release not just via pain relief, but also physiological and performance markers including band flexibility, heartrate, and cycling performance measures such as pedaling cadence, power, and record. The research design is to be commended for – as much as possible – controlling for confounding variables. This was achieved by quantifying variables of interests both before and after cycling sessions. It is of importance to note that the particular mode of self-myofascial release (SMR) used here employed a foam roller, as this is considered to be effective in massaging tissue. In brief, results indicated that the form and duration of SMR employed here did effectively reduce pain of iliotibial band friction, band flexibility, and selected performance measures including power while cycling. In general, while SMR clearly relived IT band pain while cycling, only a single measure of performance, i.e. power, was enhanced by SMR. Tis was a well-considered and planned study on a common form of pain affecting bicyclists, but there are some concerns that must be adequately addressed.
1) Be careful about repeatedly pointing out that the intervention improved cadence by 3.2% and that previous research suggests that improvements of 3.4 – 5.5 % have been deemed significant. The level here fell shy of the 3.4% cutoff value and was not fund to be statistically significant. Be careful of over-reach and focus instead on these results that were found to be significant. Also, you may want to include the P value for the 3.4% difference observed here.
2) The results reported on lines 304-307 are interesting (SMR reduced cycling record time compared to controls during the second cycling session, i.e. after receiving SMR) and deserves further consideration and attention.
3) In statistical analyses (pp 149-154) please specifically point out what the 2 main effects for ANOVA were.
4) In Results (lines 158-160) after mentioning tendency for a change in Renne’s Test results, please indicate the actual P value detected.
5) In lines 183-184, it is stated that increased pedaling power was noted among the SMR group during the second cycling session, does this reflect occurrence of pain relief?
6) It appears that pedaling duration is linked to pain remission brought on by SMR. This is an important point and deserves more attention, particularly what mechanisms may be involved, and how duration may interact with them.
Author Response
Dear Reviewer_3
Thanks for your constructive comments, and it contributed a lot to completing a more advanced manuscript. We will write a response to your comments and attach it, so please refer to it.
kind regards,
Dr. Jong-Hee Kim

Round 2
Reviewer 2 Report
All my comments have been nicely addressed.